# Bridging the Sim2Real gap with CARE: Supervised Detection Adaptation with Conditional Alignment and Reweighting

**Viraj Prabhu**[1,*]     **David Acuna**[3]     **Yuan-Hong Liao**[2]     **Rafid Mahmood**[3,4]

**Marc T. Law**[3]     **Judy Hoffman**[1]     **Sanja Fidler**[2,3]     **James Lucas**[3]

[1]*Georgia Institute of Technology*     [2]*University of Toronto*     [3]*NVIDIA*     [4]*University of Ottawa*

**Reviewed on OpenReview:** `https://openreview.net/forum?id=lAQQx7hlku`

## Abstract

Sim2Real domain adaptation (DA) research focuses on the constrained setting of adapting from a labeled synthetic source domain to an unlabeled or sparsely labeled real target domain. However, for high-stakes applications (*e.g.* autonomous driving), it is common to have a modest amount of human-labeled real data in addition to plentiful auto-labeled source data (*e.g.* from a driving simulator). We study this setting of *supervised* sim2real DA applied to 2D object detection. We propose Domain Translation via Conditional Alignment and Reweighting (`CARE`) a novel algorithm that systematically exploits target labels to explicitly close the sim2real appearance and content gaps. We present an analytical justification of our algorithm and demonstrate strong gains over competing methods on standard benchmarks.

## 1 Introduction

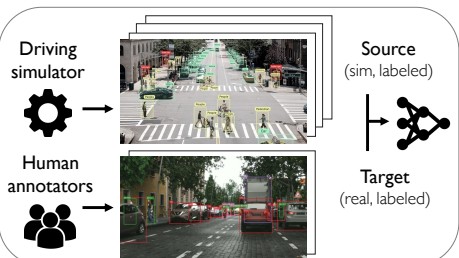

Figure 1: Traditional sim2real domain adaptation (DA) assumes access to very few or no target labels, which is unrealistic in high-stakes applications like self-driving. We study the practical setting of *supervised* Sim2Real DA applied to 2D object detection, wherein we seek to maximize target performance given human-labeled target data and an additional large set of machine-labeled simulated data.

| Method | Source labels | Target labels | mAP@50 (↑) |
|---|:---:|:---:|:---:|
| Source | ✓ | | 41.8 |
| Target | | ✓ | 62.1 |
| UDA Khindkar et al. (2022) | ✓ | | 53.1 |
| FDA Wang et al. (2020) | ✓ | ✓ | 65.2 |
| Mixing Kishore et al. (2021) | ✓ | ✓ | 64.8 |
| Seq. FT Tremblay et al. (2018) | ✓ | ✓ | 66.4 |
| Ours - CARE | ✓ | ✓ | **68.1** |

Table 1: Car detection adaptation from Sim10K (Johnson-Roberson et al., 2017) to Cityscapes (Cordts et al., 2016): combining labeled source and target data using our proposed method (`CARE`, gray row) improves over using a single data source (rows 1-2) as well as naïve combinations (rows 3-6).

Domain Adaptation (DA) is a framework that seeks to overcome shifts in data distributions between training and testing. Typically, DA methods assume access to a large amount of labeled data from the training (source) distribution, and unlabeled or sparingly labeled data from the test (target) distribution (Saenko et al., 2010; Ganin & Lempitsky, 2015; Tzeng et al., 2017). DA has been extensively studied in computer vision for applications where annotating target data is expensive (Csurka et al., 2022).

---

*Work done when while V.P. was an intern at NVIDIA. Correspondence to `virajp@gatech.edu`

As annotation costs decrease, it becomes increasingly practical to annotate more target data, especially in high-stakes industrial applications such as autonomous driving (Mahmood et al., 2022). A common practice in this field is to augment a target dataset of real driving scenarios with an additional labeled dataset generated in simulation (Karpathy, 2021; Kishore et al., 2021). Simulated data may be particularly useful to improve performance on the long-tail of driving scenarios for which it may be challenging to collect real labeled data (Rempe et al., 2022; Resnick et al., 2022). In this paper, we formulate this setting as a *supervised* Sim2Real DA problem. We use simulated, machine-labeled source data, and real, human-labeled target data (see Fig. 1), and ask: in this label-privileged setting, what would be the most effective way to combine simulated and real data to improve target performance?

Surprisingly, this practical setting has received little interest in recent domain adaptation literature, which focuses on unsupervised adaptation *(no target labels,* Chen et al. (2018); Acuna et al. (2021a); Li et al. (2022b)), and few-shot and semi-supervised adaptation (*few target labels*, Donahue et al. (2013); Wang et al. (2019a); Saito et al. (2019a); Wang et al. (2020)). Although such methods could be extended to the supervised setting, *e.g.* by adding a supervised target loss to an off-the-shelf unsupervised DA method, we find this to be sub-optimal in practice (see Table 1) since these straightforward extensions do not exploit large-scale target labels and their statistics for domain alignment. Similarly, few-shot and semi-supervised adaptation methods assume access to limited target labels (*e.g.* 8 labeled images per class for object detection, Wang et al. (2019a)) that are insufficient for reliably estimating target statistics. Facing this research gap, industry practitioners may resort to naïvely combining labeled source and target data via mixing (Kishore et al., 2021) (*i.e.* training on combined source and target data) or sequential fine-tuning (Tremblay et al., 2018; Prakash et al., 2019; 2021) (*i.e.* training on source data followed by finetuning on target data). However, these simple heuristics do not address the domain gap between simulation and reality.

This paper addresses the research-practice gap to show that *systematically* combining the two labeled data sets can significantly improve performance over competing methods (see Table 1). We propose a general framework called *Domain Translation via Conditional Alignment and Reweighting* (`CARE`) for supervised Sim2Real DA. `CARE` builds on commonly-used baselines and off-the-shelf adaptation methods but explicitly leverages existing labels in the target domain to minimize both appearance gaps (pixel and instance-level visual disparity) and content gaps (disparities in task label distributions and scene layout). Specifically, we overcome the appearance gap by explicitly using ground-truth labels to conditionally align intermediate instance representations. To overcome the content gap, we conditionally reweight the importance of samples using estimated spatial, size, and categorical distributions. We formalize our setting using the joint risk minimization framework and provide theoretical insights for our design choices. Finally, we apply our framework to the challenging task of 2D object detection. We make the following contributions:

(1) We present the first detailed study of supervised Sim2Real object detection adaptation, addressing a large research-vs-practice gap between unsupervised and few-shot domain adaptation with an industry-standard practice of combining labeled data from simulated and real domains.

(2) We show that existing adaptation methods yield sub-optimal performance in this setting by not adequately exploiting target labels. We propose `CARE`, an algorithm that systematically exploits target labels to bridge the sim2real domain gap by performing conditional alignment and reweighting. On three Sim2Real benchmarks for detection adaptation, `CARE` strongly outperforms competing methods (*e.g.* boosting mAP@50 by as much as ∼25% on Synscapes→Cityscapes).

(3) We formalize our setting using the joint risk minimization framework and provide theoretical insights into our design choices.

## 2 Related work

To our knowledge, supervised domain adaptation (SDA) for object detection has not seen recent work in computer vision. Early DA works (Saenko et al., 2010; Kulis et al., 2011; Hoffman et al., 2013; Tsai et al., 2016) have studied the SDA setting applied to image classification, proposing contrastive-style approaches based on metric learning with cross-domain pairwise constraints. However, these works predate deep learning

and do not study complex tasks like object detection. Below, we summarize lines of work in the related areas of unsupervised and few-shot adaptation.

**Unsupervised domain adaptation (UDA)**. The DA literature primarily focuses on *unsupervised* adaptation from a labeled source setting to an unlabeled target domain (Saenko et al., 2010; Ganin & Lempitsky, 2015; Hoffman et al., 2018; Zhang et al., 2020). Successful UDA approaches have employed different strategies ranging from domain adversarial learning Long et al. (2015); Acuna et al. (2021b) to domain discrepancy minimization (Long et al., 2018), image translation (Hoffman et al., 2018), and self-training (Prabhu et al., 2021; Li et al., 2022b). Cross-domain object detection has also seen recent work, based on multi-level domain adversarial learning (Chen et al., 2018), strong-weak distribution alignment of local and global features (Saito et al., 2019b), and domain adversarial learning weighted by region discriminativeness (Zhu et al., 2019), Alternatively, RoyChowdhury et al. (2019); Li et al. (2022b) self-train with refined pseudo-labels, and Kim et al. (2019) use background regularization.

Importantly, due to the absence of target labels, UDA methods resort to approximations based on marginal alignment or pseudo-labels. In this paper, we instead consider *supervised* sim2real adaptation where ground-truth labels are provided for the target dataset during training. To compare against our approach, we benchmark supervised extensions of existing UDA methods as baselines in our paper. While our setting bears similarity to general transfer learning (Zhai et al., 2019), it still makes the additional standard DA assumption of (large to full) label set overlap between train and test distributions. As a result, in DA such transfer can be realized by directly aligning class features across domains.

**Few-shot (FDA) and Semi-supervised Domain Adaptation (SSDA).** Closer to our setting are Few-shot DA learning (FDA, Wang et al. (2019a); Gao et al. (2022); Zhong et al. (2022); Ramamonjison et al. (2021)) and Semi-supervised DA (SSDA, Donahue et al. (2013); Yao et al. (2015); Saito et al. (2019a)), which differ in important ways. FDA assumes a very small amount of labeled target data is available (*e.g.* 8 images per class for detection in Wang et al. (2019a)). Such methods employ source feature-regularization with instance-level adversarial learning (Wang et al., 2019a), point-wise distribution alignment (Zhong et al., 2022), and multi-level domain-aware data augmentation (Gao et al., 2022). SSDA also assumes limited target labels (*e.g.* 1 to 3 images per category for image classification (Saito et al., 2019a)), but additionally leverages a large set of *unlabeled* target data, making use of min-max entropy optimization (Saito et al., 2019a) or student-teacher learning frameworks (Li et al., 2022b). Closest to our work is Motiian et al. (2017), which also considers supervised adaptation of image classifiers by minimizing appearance gap via cross-domain semantic alignment, focusing on the setting wherein target data is sparingly labeled (*e.g.* 3 examples/class on Office-31 (Saenko et al., 2010)). In contrast, we study supervised DA in the context of object detection, which introduces an additional *content* gap arising from changes in scene layout and task label distributions. However, we assume access to a substantial amount of labeled target data in addition to a large (in theory, possibly infinite) amount of labeled simulated data. This assumption uniquely permits *reliable* estimation of target statistics. Our algorithm leverages these statistics and target labels to systematically close the sim2real domain gap.

## 3 Approach

In this section, we first introduce the supervised Sim2Real detection adaptation problem (Section 3.1). We characterize two primary aspects of the Sim2Real domain gap: an appearance gap and a content gap (Section 3.2). Finally, we introduce our method, `CARE`, that leverages a labeled target dataset to close this domain gap (Section 3.3) and provide an analytical justification of the algorithm (Section 3.4).

### 3.1 Problem Formulation

Let $\mathcal{X}$ and $\mathcal{Y}$ denote input and output spaces. In object detection, $x \in \mathcal{X}$ are images ($\mathcal{X} \subseteq \mathbb{R}^{H \times W \times 3}$) and $y := (B, C) \in \mathcal{Y}$ are $K$-class labels with $C \in \{1, .., K\}$ and bounding boxes $B \subseteq \{(\mathsf{w}, \mathsf{h}, \mathsf{x}, \mathsf{y}) \in \mathbb{R}^4\}$ (comprising the width $\mathsf{w}$, height $\mathsf{h}$, and centre coordinates $(\mathsf{x}, \mathsf{y})$, respectively). Let $h(x) := h_\theta(g_\phi(x))$ be an object detector composed of a feature extractor $g(x)$ and a classifier $h(g(x))$ that are parameterized by $\phi$ and $\theta$, respectively. Matching prior object detection work (Khindkar et al., 2022; Wang et al., 2021), we

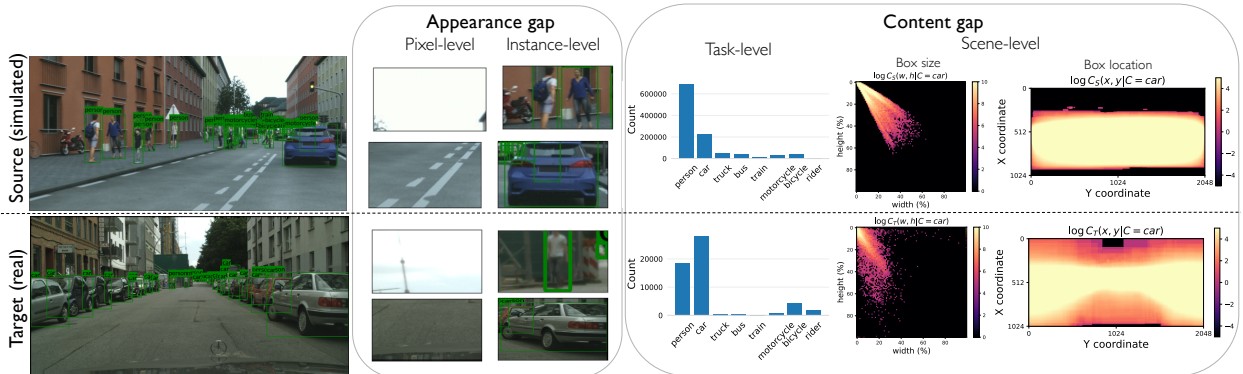

Figure 2: The domain gap between a simulated source and a real target domain can be decomposed into an appearance gap and a content gap. The appearance gap corresponds to pixel-level differences (*e.g.* texture and lighting) and instance-level differences (*e.g.* vehicle design). The content gap consists of differences in label distributions due to different class frequencies and bounding box sizes and locations. **Right.** Column 1: Task label histograms. Column 2: Empirical distribution of "car" box *sizes*. Column 3: Empirical distribution of "car" box *locations*.

design $h(g(x))$ via Faster R-CNN (Ren et al., 2015), which uses a region proposal network that receives features generated by a backbone network and passes them through a Region of interest (ROI) align layer to obtain ROI features; these are then passed through a final box predictor. We let $\hat{B}, \hat{C} = \arg\max h(g(x))$ be bounding box coordinates and object class predicted by the model for input image $x$. In sim2real SDA, we are given two labeled data sets representing a (simulated) source distribution $P_S$ and a (real) target distribution $P_T$. Our goal is to minimize the expected risk of a detection loss consisting of a classification loss $\ell_{cls}$ and bounding box regression loss $\ell_{box}$:

$$\ell_{det}(h(g(x)), B, C) := \ell_{box}(\hat{B}, B) + \ell_{cls}(\hat{C}, C) \tag{1}$$

over a target domain $r_T := \mathbb{E}_{x,B,C \sim P_T}[\ell_{det}(h(x), B, C)]$.

## 3.2 Characterizing the Sim2Real Domain Gap

Leveraging the source distribution to improve performance on the target is challenging due to the *domain gap* which exists in both the image and label distributions. We partition this gap into two categories: appearance gap and content gap (Kar et al., 2019) and characterize these in detail, using the Synscapes (Wrenninge & Unger, 2018)→ Cityscapes (Cordts et al., 2016) shift for object detection adaptation as an example.

The **appearance gap** consists of visual disparities between images from the two domains (see Fig 2, *left*). For example, a pixel-level appearance gap may be due to differences in lighting between real and simulated images (Chattopadhyay et al., 2022), while an instance-level gap may be due to differences in the appearance of synthesized versus real objects. We characterize the appearance gap as the dissimilarity $D(\cdot, \cdot)$ in the probabilities between source and target distributions when conditioned on the label (*e.g.* $D(P_S(x|B, C), P_T(x|B, C))$).

The **content gap** can be decomposed into scene-level changes in the layout of objects (*e.g.* size and spatial distribution) as well as shifts in the task label distributions and the frequencies of classes (see Fig 2, *right*). We characterize the scene-level changes as the dissimilarity in the probabilities of object bounding boxes when conditioned on the class $D(P_S(B|C), P_T(B|C))$ and the task-level class frequency gap as the dissimilarity in class probabilities $D(P_S(C), P_T(C))$.

## 3.3 Bridging the domain gap with CARE

To close the sim2real gap, *Conditional Alignment and Reweighting* (CARE) minimizes the effect of both the appearance and the content gaps via feature alignment and importance reweighing. Let $w_S(C) := 1/P_S(C), w_T(C) := 1/P_T(C)$ be the inverse class frequency for each domain and let $v(B|C) :=$

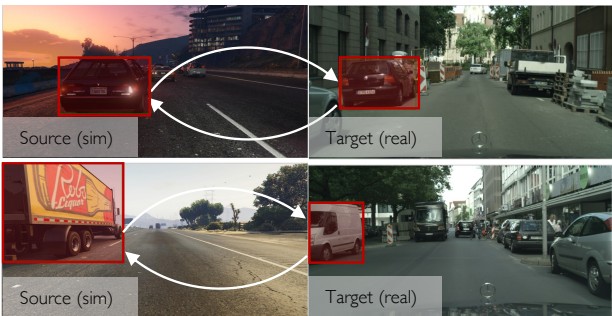

Figure 3: Visualization of cross-domain cycle consistency matching with `CARE` on Sim10K→Cityscapes. `CARE` embeds similar-looking cars closer to minimize the appearance gap.

$P_T(B|C)/P_S(B|C)$ be the inverse ratio of the scene-level bounding box frequency gap. These reweighting factors ensure that the learned classifier considers that the source and target data sets follow the same distribution during training. In `CARE`, we minimize the following *domain translation* loss:

$$
\begin{aligned}
\min_{\theta,\phi} \; & \mathbb{E}_{x,B,C\sim P_S}\left[w_S(C)v(B|C)\ell_{det}(h(g(x)),B,C)\right] \\
& + \mathbb{E}_{x',B',C'\sim P_T}\left[w_T(C')\ell_{det}(h(g(x')),B',C')\right] \\
& + \lambda\mathbb{E}_{\substack{x',B',C'\sim P_T \\ x,B,C\sim P_S}}\left[\ell_{align}(g(x),g(x'))\Big|C=C'\right].
\end{aligned}
\tag{2}
$$

where $\ell_{align}$ is defined in Equation 3, and $\lambda \geq 0$ is a regularization parameter. The above loss minimizes three terms, where the first term is a reweighted detection loss over the source dataset and the second loss is a class-balanced detection loss over the target dataset. The third term aligns the encoded features $g(x)$ and $g(x')$ of similar cross-domain instance embeddings belonging to the same class. We now elaborate upon each.

### 3.3.1 Bridging appearance gap with cross-domain cycle consistency

To minimize the appearance gap, $\ell_{align}$ performs a class-and-box conditional feature alignment strategy by optimizing a cross-domain cycle consistency objective. Specifically, we extract ROI features corresponding to the ground truth bounding box coordinates of both source and target images and match similar cross-domain instance features belonging to the same class. Fig. 3 illustrates this intuition.

**Matching source to target.** For a given class, suppose we are given $k$ ground truth bounding boxes from the source and target domains each. For each instance, our encoder extracts $d$-dimensional ROI features: let $\mathbf{u}^i \in \mathbb{R}^d$ and $\mathbf{v}^j \in \mathbb{R}^d$ denote features extracted from the $i$-th source and $j$-th target instances. For source instance $i$, we compute *soft-matching* target features $\hat{\mathbf{v}}^i$ as an average of target features $\mathbf{v}^j$ *weighted* by their similarity to $\mathbf{u}^i$. To measure pairwise cross-domain similarity $s_{i,j}$, we employ negative squared Euclidean distance between features. Such soft-matching has been shown to be more stable than matching to a single target instance (Dwibedi et al., 2019; Wang et al., 2019b).

$$
\hat{\mathbf{v}}^i := \sum_{j=1}^{k} \alpha_{i,j}\mathbf{v}^j, \; \text{where} \; \alpha_{i,j} := \frac{e^{s_{i,j}}}{\sum_{m=1}^{k} e^{s_{i,m}}} \; \text{and} \; s_{i,j} := -\|\mathbf{u}^i - \mathbf{v}^j\|_2^2.
$$

**Matching back to source.** Next, we form a cross-domain cycle by matching soft target features back to the source. As before, we first compute pairwise similarity scores for backward matching $\hat{s}_{m,i} := -\|\mathbf{u}^m - \hat{\mathbf{v}}^i\|_2^2$, and assemble a similarity score vector $\mathbf{s}^i := [\hat{s}_{1,i},\ldots,\hat{s}_{m,i}\ldots,\hat{s}_{k,i}] \in \mathbb{R}^k$. We treat this vector as logits and minimize the following cross-entropy loss:

$$
\ell_{align}(\mathbf{u},\hat{\mathbf{v}}^i) := \sum_{m=1}^{k} \mathbb{1}_{m=i}\left(\log\left(\texttt{softmax}(\mathbf{s}^i)_m\right)\right) \; \text{where} \; \mathbb{1}_{m=i} \; \text{is the indicator function.}
\tag{3}
$$

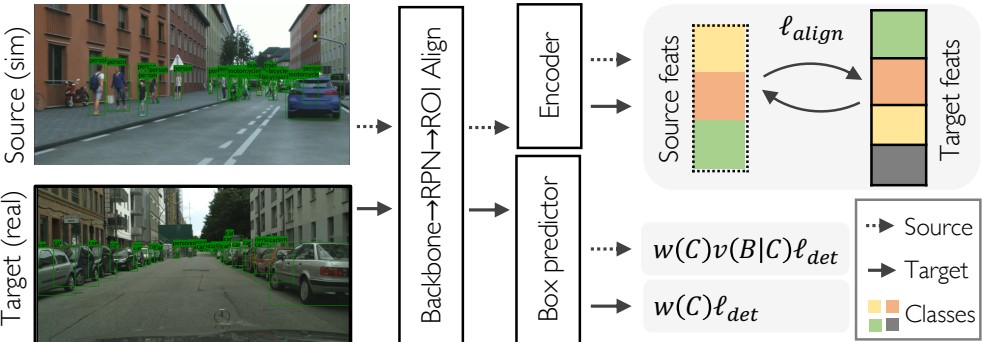

Figure 4: Conditional Alignment and Reweighting (`CARE`) exploits target labels to estimate and bridge cross-domain appearance gaps (via a cycle consistency-based conditional feature alignment objective) and content gaps (via importance reweighting).

Intuitively, this self-supervised cycle consistency objective encourages aligning feature representations for visually similar cross-domain instances belonging to the same class. Further, in the supervised DA setting, access to labels in both source and targets allows us to condition on ROI features corresponding to ground-truth (rather than proposed) bounding boxes, which ensures that the features being aligned are well-localized.

The above approach is inspired by a temporal cycle confusion objective proposed for robust object detection from videos (Wang et al., 2021). However, we make several fundamental modifications: (i) `CARE` matches instances across domains (rather than time), (ii) `CARE` matches instances from the same class, whereas Wang et al. (2021) study a 1 class setting, (iii) `CARE` aligns features corresponding to ground truth (rather than predicted) ground truth boxes, and (iv) we match instances that are close (rather than far) in feature space. Of these, (i) is necessary for use in domain adaptation, whereas (ii-iv) are additional changes we propose to simplify learning and account for the large visual discrepancy across classes and domains. In Section 4.5, we demonstrate that our modifications improve performance. Our alignment loss reduces to the classification of the soft nearest neighbors and therefore tends to be robust to label noise (Dwibedi et al., 2019).

We remark here that despite these novel modifications, our primary contribution is actually a *theoretically-motivated joint framework* for supervised domain adaptation that combines conditional alignment and importance reweighting and is agnostic to the underlying feature alignment strategy. To validate this, in Section 4.5 we present adaptation results by swapping out only the feature alignment strategy from cycle consistency to class-and-box-conditional versions of two popularly used domain adaptation strategies and demonstrate consistent improvements.

### 3.3.2 Bridging content gap with importance reweighting

To close the task label distribution content gap, we apply inverse frequency reweighing to simulate a balanced label distribution in the source and target domains. For each domain $\omega \in \{S, T\}$, we reweigh instances of class $C$ via multiplicative class weights $w_\omega(C) \propto 1/N_\omega(C)$, where $N_\omega(C)$ is the number of training examples in domain $\omega$.

We approximate the class-conditional box ratios as follows:

$$\frac{P_T(B|C)}{P_S(B|C)} \approx \frac{P_T(\mathsf{w}, \mathsf{h}|C)}{P_S(\mathsf{w}, \mathsf{h}|C)} \frac{P_T(\mathsf{x}, \mathsf{y}|C)}{P_S(\mathsf{x}, \mathsf{y}|C)} =: v(B|C) \tag{4}$$

Intuitively, this ratio up-weights boxes of a class that are of a size and location relatively more represented in the target than in the source. Note that the first approximate equality $\approx$ is due to an assumption of independence between $(\mathsf{w}, \mathsf{h})$ and $(\mathsf{x}, \mathsf{y})$, which we assume to simplify computations. We estimate each probability component via class-conditional Gaussian kernel density estimation (KDE) (Scott, 2015) fitted to the ground truth bounding box locations and sizes respectively. In Appendix A.2, we include details of this estimation, including appropriate smoothing and thresholding to handle regions with low target support.

Note that while class-rebalancing is well-studied for class-imbalanced problems, to our knowledge, we are the first to propose class-conditional box reweighting for object detection adaptation to mitigate a sim2real content gap. In Section 4.5 we demonstrate that such reweighting consistently improves performance.

### 3.4 Analytical justification

We now analyze our loss function in Equation 2 to develop a theoretical intuition for its effectiveness. Let us rewrite the first term in the loss as follows:

$$\mathbb{E}_{P_S}\left[w_S(C)v(B|C)\ell_{det}(h(g(x)),B,C)\right] = \mathbb{E}_{P_T}\left[\frac{P_S(x,B,C)}{P_T(x,B,C)}w_S(C)v(B|C)\ell_{det}(h(g(x)),B,C)\right] \quad (5)$$

$$= \mathbb{E}_{P_T}\left[\frac{P_S(C)}{P_T(C)} \times \frac{P_S(B|C)}{P_T(B|C)} \times \frac{P_S(x|B,C)}{P_T(x|B,C)} \times w_S(C)v(B|C)\ell_{det}(h(g(x)),B,C)\right]. \quad (6)$$

Above, Equation 5 follows from importance reweighting, and Equation 6 follows from Bayes rule. Next, recall that $w_S(C) = 1/P_S(C)$ and $v(B|C) \approx P_T(B|C)/P_S(B|C)$. Substituting these two, we obtain

$$\text{Eq. } 6 \approx \mathbb{E}_{P_T}\left[\frac{P_S(x|B,C)}{P_T(x|B,C)}\frac{1}{P_T(C)}\ell_{det}(h(g(x)),B,C)\right] \quad (7)$$

Finally, recall our feature alignment component, which is designed to minimize the distance between encoded features of the same class and box statistics. Successfully minimizing the third term in Equation 2 should obtain $P_S(g(x)|B,C) = P_T(g(x)|B,C)$. Using this, we obtain

$$\text{Eq. } 7 \approx \mathbb{E}_{P_T}\left[\frac{P_S(g(x)|B,C)}{P_T(g(x)|B,C)}\frac{1}{P_T(C)}\ell_{det}(h(g(x)),B,C)\right]$$

$$= \mathbb{E}_{P_T}\left[\frac{1}{P_T(C)}\ell_{det}(h(g(x)),B,C)\right] \quad (8)$$

where the first line follows from the assumption that feature-level distances should reflect image appearance distances, and the second line follows from minimizing $\ell_{align}$. Overall, Equation 8 and the second term in Equation 2 minimize a class-weighted version of the expected risk $r_T$. In our case, the target metric is mean AP, which values performance on all classes equally. Since in practice target distributions often feature imbalanced classes, this modified risk simulates a balanced label distribution and better maximizes mAP.

The steps above follow from several assumptions including independence of box position and size, equivalence between the ratios of feature-level probabilities and appearance probabilities, and that the target support is a subset of that of the source. Further, it relies on successfully minimizing this feature-level gap. Nonetheless, as we show in the next section, our method demonstrates powerful empirical performance in the target domain.

## 4 Results

We now describe our experimental setup for object detection adaptation: datasets and metrics (Section 4.1), implementation details (Section 4.2), and baselines (Section 4.3). We then present our results (Section 4.4) and ablate (Section 4.5) and analyze our approach (Section 4.6). Our experiments reveal that CARE consistently outperforms alternatives including training with only target data, naïve strategies for combining data sets, as well as modifications of vanilla UDA strategies. Moreover, these improvements are consistent across classes, box sizes, and error types. Finally, our strong performance necessitates both the feature alignment and the class rebalancing techniques combined.

### 4.1 Datasets and metrics

We perform domain adaptation from three different source data sets of synthetic images, Sim10K (Johnson-Roberson et al., 2017), Synscapes (Wrenninge & Unger, 2018), and an internal data set we denote by

Internal-Sim. Sim10K contains 10,000 images of 1914×1052 resolution with pixel-level annotations extracted from the game GTA-5. Synscapes is a photo-realistic dataset of 25,000 synthetic driving scenes of 1440×720 resolution. Finally, Internal-Sim is a private synthetic data set of 48,000 photo-realistic driving scenes. Synscapes and Internal-Sim exhibit a long-tailed category distribution (see Fig. 2). For each source, we train an object detector to adapt to our target, Cityscapes (Cordts et al., 2016) which is a data set of 2,500 real driving images.

For our primary evaluations, we use 25% of labeled target data (625 images on Cityscapes) and the entire labeled source set (10k/25k/48k respectively), to mirror the practical scenario of having access to a modest amount of labeled real data in addition to an order of magnitude more labeled simulated data. We find that this regime is where the conventional practitioner baselines of mixing and sequential fine-tuning show the largest improvement. On the other hand, if the target data set size is as large as the source size, we observe minimal improvements with conventional strategies to be minimal. While we use this as our primary evaluation setting to compare against baselines, we further validate our strategy on the full ranges of target data to get a complete analysis; we find that using `CARE` has a large positive impact over all baselines no matter how much source versus target data is available (see Fig 5). For **Sim10K→Cityscapes**, we focus on object detection for a single class (*i.e.* car) to better compare against prior Sim2Real domain adaptation methods (Khindkar et al., 2022). For **Synscapes→Cityscapes** and **Internal-Sim→CityScapes**, we evaluate object detection for the subset of classes that are shared between source and target, corresponding to eight and three classes respectively. To evaluate all models, we match prior work Chen et al. (2018); Khindkar et al. (2022); Wang et al. (2021) and report per-category Average Precision (AP) and its mean across classes at an IoU threshold of 50% (mAP@50), over the target test set.

### 4.2 Implementation details

We use a Faster-RCNN Ren et al. (2015) architecture with a ResNet-50 He et al. (2016) backbone. We run 10k iterations of SGD with a learning rate of 0.01, momentum of 0.9, weight decay of $10^{-4}$, and learning rate warmup matching Wang et al. (2021). We set $\lambda = 0.1$ in Equation 2. We use 8 NVIDIA V100 GPUs with a per-GPU batch size of 4 and maintain a 1:1 within-batch source-to-target ratio across experiments. For more details see the appendix.

### 4.3 Baselines

We compare the performance of `CARE` to a diverse range of baselines designed for the unsupervised, few-shot, and supervised DA regimes, in addition to proposing two *new* baselines for supervised DA.

**Control.** As controls, we benchmark the following single-domain baselines:

(1) **Source only**: Supervised learning using only the labeled source dataset.

(2) **Target only**: Supervised learning using only the labeled target dataset.

**Unsupervised DA.** We compare against a recently proposed method that only uses labeled source and unlabeled target data:

(3) **ILLUME** (Khindkar et al., 2022): For completeness, we copy results on Sim10K→Cityscapes of a state-of-the-art UDA method that uses labeled source and unlabeled target data.

**Few-shot DA.** We compare against a recently proposed method designed for use in the labeled source and few-shot target regime:

(4) **TFA** (Wang et al., 2020): TFA is a two-stage finetuning algorithm proposed for few-shot object detection that updates all parameters on source (base) data followed by finetuning only the final layer (box regressor and classifier) on a balanced dataset of source and target data. However, we observe relatively low performance with finetuning only the last layer (despite using a lower learning rate as recommended and both with and without weight re-initialization). Instead, we report results *without* freezing weights in the second phase. Note that we provide TFA with an identical amount of labeled target data as our method for a fair comparison.

**Supervised DA.** We benchmark the following supervised DA baselines:

Table 2: Results for supervised sim2real object detection adaptation on target. We compare CARE to source and target only training, a state-of-the-art unsupervised DA method (ILLUME Khindkar et al. (2022)), naïve sim+real combinations (mixing Kishore et al. (2021) and sequential finetuning Tremblay et al. (2018)), supervised extensions of popular UDA methods (DANN Ganin & Lempitsky (2015) and MMD Long et al. (2015)),and a recently proposed few-shot detection strategy Wang et al. (2020).

| Method | mAP@50 (↑) |
|---|---|
| Source | 41.8 |
| UDA | 53.1 |
| Target | 62.1 |
| Mixing | 64.8 |
| Seq. FT | 66.4 |
| S-MMD | 65.8 |
| S-DANN | 65.3 |
| FDA | 65.2 |
| CARE (ours) | **68.1** |

(a) Sim10K→Cityscapes (1-way)

| Method | mAP@50 (↑) |
|---|---|
| Source | 19.2 |
| Target | 34.2 |
| Mixing | 39.0 |
| Seq. FT | 39.8 |
| S-MMD | 40.0 |
| S-DANN | 40.8 |
| CARE (ours) | **48.5** |

(b) Synscapes→Cityscapes (8-way)

| Method | mAP@50 (↑) |
|---|---|
| Source | 22.5 |
| Target | 45.2 |
| Mixing | 49.3 |
| Seq. FT | 45.4 |
| S-MMD | 50.6 |
| S-DANN | 49.8 |
| CARE (ours) | **53.7** |

(c) Internal-Sim→CityScapes (3-way)

(5) **Mixing** (Kishore et al., 2021): Supervised learning on the combined source and target data sets, while maintaining a 1:1 ratio within batches (we ablate this mixing ratio in appendix).

(6) **Sequential Finetuning** (Tremblay et al., 2018): Supervised learning on the source dataset followed by finetuning all layers of the model with the target dataset.

We also propose and benchmark supervised extensions of two popular UDA strategies. Similar to CARE, both methods first extract ROI features corresponding to ground truth bounding box coordinates for all source and target instances within a batch, but differ in their feature alignment objective:

(7) **S-MMD**: A class and box-conditional *supervised* version of Maximum Mean Discrepancy (Long et al., 2015). S-MMD computes the per-class cross-domain maximum mean discrepancy (MMD) in feature space and then minimizes its average across classes as a domain alignment loss using a linear kernel.

(8) **S-DANN**: A class and box-conditional *supervised* version of DANN (Ganin & Lempitsky, 2015). S-DANN trains a domain classifier to distinguish between per-class source and target box features and optimizes a domain adversarial loss that updates the feature extractor to fool this domain classifier between cross-domain box features corresponding to the same class, similar to Chen et al. (2018).

### 4.4 Main Results

Table 2 summarizes our results. We find:

▷ **Combining simulated and labeled real data improves performance.** We first confirm that supervised learning using only the target data outperforms the setting of using only source data (*e.g.* **+20.3** on Sim10K→Cityscapes). Further, across all shifts, supervised baselines that naïvely combine simulated and real data (*i.e.* mixing and sequential finetuning) outperform training using only the target data. This shows that additional simulated data is helpful. Interestingly, we find that sequential finetuning outperforms mixing on two of three shifts.

▷ **CARE outperforms competing methods for unsupervised DA and few-shot DA.** First, we note on Sim10K→Cityscapes, CARE strongly outperforms state-of-the-art methods for unsupervised DA (Khindkar et al., 2022) that use only labeled source and unlabeled target data (**+14.4**), and few-shot DA (Wang et al., 2020) that use labeled source and target data[1](**+2.9**).

---

[1]For fair comparison, we assume that both FDA and CARE access the exact same amount of labeled source and target data.

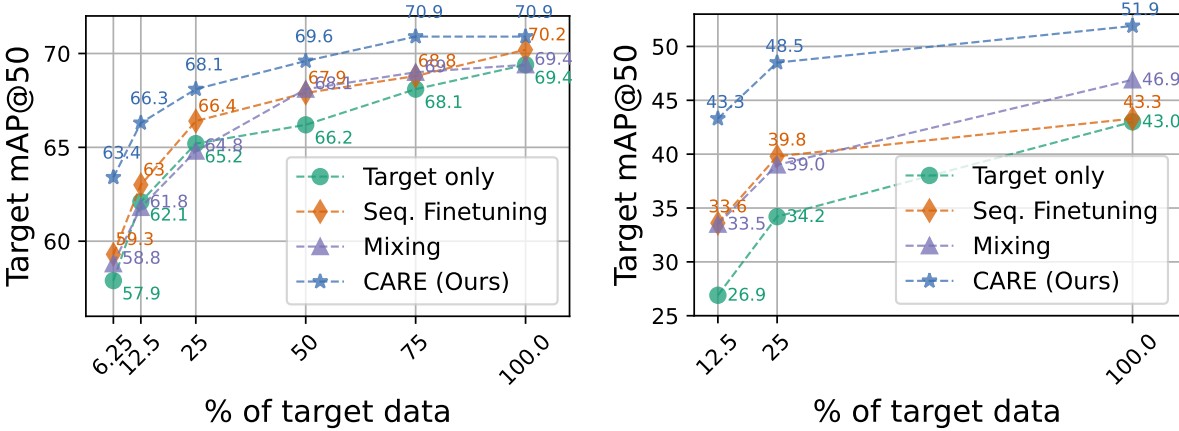

Figure 5: Plotting the scaling behavior of various methods using 100% of labeled source data and a varying amount of labeled target data, on Sim10K→Cityscapes (*left*) and Synscapes→Cityscapes (*right*). CARE consistently dominates the baselines across settings, often by significant margins (**+3.3** and **+9.7** mAP@50 over the next best method with 12.5% of the target domain labeled).

| Method | FRCNN w/ Swin-T | VitDet |
|---|---|---|
| Target | 44.8 | 42.6 |
| Mixing | 44.9 | 37.9 |
| CARE (Ours) | **53.5** | **54.9** |

Table 3: Target mAP@50 for target only training, mixing, and CARE on 8-way Synscapes→Cityscapes adaptation with i) Faster-RCNN (Ren et al., 2015) with Swin-T (Liu et al., 2021) backbone and ii) ViTDet (Li et al., 2022a).

▷ **CARE outperforms naïve strategies that combine source and target data.** Finally, we find that across all shifts, CARE outperforms strategies based on based on mixing (Kishore et al., 2021) (**+3.3, +9.5, +4.4** mAP@50) and sequential finetuning (Tremblay et al., 2018) (**+1.7, +8.7, +8.3** mAP@50). This suggests that the Sim2Real domain gap is a barrier to effective mixing, and systematically mitigating it using target labels is beneficial. Most importantly, we outperform each benchmarked supervised extension of UDA on all shifts. This result validates the research-practice gap by showing that UDA cannot be easily extended to the practical setting of labeled target data, thereby necessitating CARE in supervised domain adaptation.

▷ **Gains with CARE are consistent across target labeling budgets.** While we focus on the setting wherein a modest amount of target data is labeled (*e.g.* 25%=625 images on Cityscapes), in addition to a large amount of labeled source data (10K images from Sim10K) a natural question that arises is the versatility of CARE across varying amounts of labeled target data. To study this, we benchmark the performance of supervised DA strategies as we vary the amount of labeled target data available (and 100% of source data wherever applicable). Figure 5 shows results on the Sim10K→Cityscapes (*left*) and Synscapes→Cityscapes (*right*). As seen, CARE *consistently* dominates competing methods across labeling budgets, with gains being particularly significant in the low target data regime (**+3.3** and **+9.7** mAP@50 over the next best method with 12.5% of the target domain labeled). More generally, we find that methods that combine simulated and real data show strong improvements over target-only training in the low-target data regime and diminishing returns thereafter. [2]

▷ **Gains with CARE are consistent across backbones.** In addition to the standard Faster R-CNN with a ResNet-50 backbone that we employ across detection experiments for fair comparison to prior work (Chen et al., 2018; Wang et al., 2021; Li et al., 2022b), we now test generality by running CARE with two additional

---

[2]We note that on Sim10K→Cityscapes, even performance with target-only training saturates at higher target label budgets, and so it is unclear whether the diminishing returns are due to performance saturation rather than the reduced utility of simulation.

Table 4: Ablating `CARE` on all three shifts. Our method is in gray with the improvement versus mixing in small font.

| # | $P(g(x)|B,C)$ alignment | $P(C)$ rewt. | $P(B|C)$ rewt. | mAP@50 ($\uparrow$) Sim10k | Synscapes | Internal-Sim |
|---|---|---|---|---|---|---|
| 1 | (Mixing baseline) | | | 64.8 | 39.0 | 49.3 |
| 2 | S-MMD | | | 65.8 | 40.0 | 50.6 |
| 3 | S-DANN | | | 65.3 | 40.8 | 49.8 |
| 4 | Cycle Consistency | | | 67.2 | 41.8 | 50.8 |
| 5 | None (Mixing baseline) | ✓ | | 64.8 | 46.1 | 51.8 |
| 6 | Cycle Consistency | ✓ | | 67.2 | 46.6 | 52.5 |
| 7 | Cycle Consistency | ✓ | ✓ | $\mathbf{68.1}_{+3.3}$ | $\mathbf{48.5}_{+9.5}$ | $\mathbf{53.7}_{+4.4}$ |

Table 5: Ablating our proposed conditional reweighting strategies on Synscapes → Cityscapes.

| Method | w/o CB | w/ CB |
|---|---|---|
| Source | 19.2 | 20.0 |
| Target | 34.2 | 40.0 |
| Mixing | 39.0 | 46.1 |
| Seq. FT | 39.8 | 44.9 |

| Method | mAP@50 |
|---|---|
| $P(\mathsf{w},\mathsf{h},\mathsf{x},\mathsf{y}|C)$ | 48.5 |
| Only $P(\mathsf{x},\mathsf{y}|C)$ | 46.7 |
| Only $P(\mathsf{w},\mathsf{h}|C)$ | 48.3 |
| None | 46.6 |

(a) Ablating $P(C)$ reweighting.

(b) Ablating $P(B|C)$ reweighting.

architectures: **(i)** Replacing the ResNet-50 (He et al., 2016) backbone in Faster R-CNN (Ren et al., 2015) to Swin-T (Liu et al., 2021): Swin is a hierarchical vision transformer architecture that uses shifted window attention across non-overlapping local windows which limits computation complexity to be linear with respect to image size, which improves efficiency and in turn allows for modeling multi-scale imagery. **(ii)** VitDet (Li et al., 2022a): ViTDet employs a plain, non-hierarchical vision transformer backbone (initialized with MAE (He et al., 2022) pretraining) with a simple feature pyramid network and standard window attention with a few cross-window propagation blocks. Despite its simple design, ViTDet achieves strong detection performance.

Note that both of these make use of Vision Transformer Dosovitskiy et al. (2020) backbones, and ViTDet additionally benefits from self-supervised pretraining with MAE He et al. (2022). We report target mAP@50 for target-only training, mixing, and CARE on 8-way Synscapes→Cityscapes adaptation. As seen, even with recent vision transformer-based backbones and detectors CARE provides consistent and strong improvements over the baseline (eg. absolute gain of $+$**8.6** with Swin-T, $+$**17** with VitDet, over mixing).

### 4.5 Ablation study

In Table 4, we ablate the various components of `CARE`. We find that:

▷ **Class-and-box conditional feature alignment is necessary (Rows 2-4 vs. 1).** Regardless of the specific feature alignment strategy (*i.e.* S-MMD, S-DANN, and our proposed cross-domain Cycle Consistency), additional feature alignment improves performance.

We also remark that during model design, we tested variations of Cycle Consistency-based alignment on Sim10K→Cityscapes by i) conditioning on *predicted* rather than ground truth class and box coordinates (66.1 mAP@50, **-1.1** compared to our method), and ii) conditioning on predicted box coordinates and ignoring class predictions (64.9 mAP@50, roughly on par with mixing). These two settings yielded 66.1 mAP@50 (**-1.1** versus Row 4) and 64.9 mAP@50 (**-2.3** versus Row 4), respectively.

▷ **Class rebalancing with $P(C)$ reweighting is highly effective (Row 5 vs. 1).** Particularly on multi-class source settings (*e.g.* Synscapes and Internal-Sim), $P(C)$ reweighting considerably boosts performance

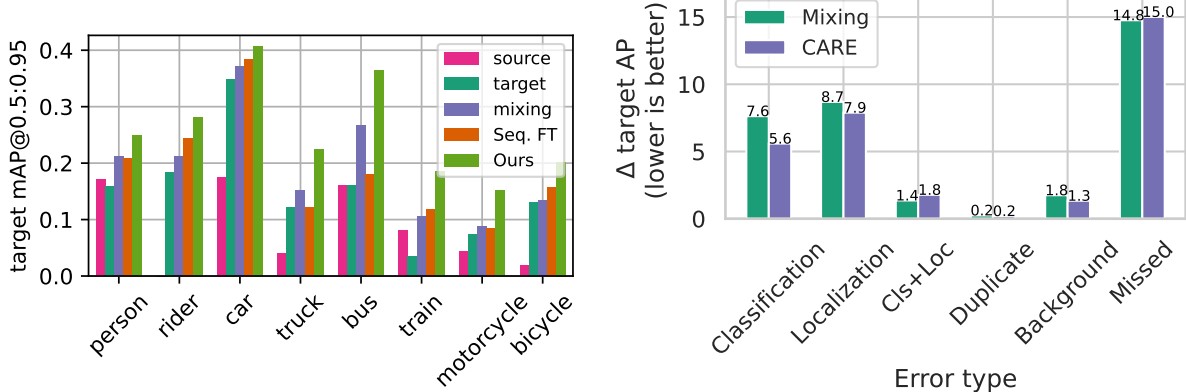

Figure 6: **(left)** Per-class performance comparison of `CARE` to baselines on Synscapes→Cityscapes. **(right)** Visualizing change in dAP (lower is better) Bolya et al. (2020) for errors of different types using `CARE`, over a mixing baseline.

(*e.g.* by **+7.1** mAP@50 on Synscapes→Cityscapes). Further, Table 5 (a) shows that class balancing naturally improves the baselines as well, due to mAP evaluating classes equally.

▷ **Importance reweighting based on estimated $P(B|C)$ statistics further improves performance (Row 7 vs. 6).** Finally, we show that our novel proposal of additional class-conditional box reweighting consistently improves performance across all shifts. Table 5 (b) presents results for different formulations of $P(B|C)$. It validates our reweighting scheme which decomposes box size with $P(\mathsf{w},\mathsf{h}|C)$ and location with $P(\mathsf{x},\mathsf{y}|C)$. Capturing both is better than using only one (**+1.8, +0.2** mAP@50) or neither (**+1.9** mAP@50).

▷ **Our proposed feature alignment objective outperforms prior work.** Recall that we propose to align similar rather than dissimilar cross-domain instances belonging to the same class using a cycle consistency objective, unlike prior work that uses cycle confusion (Wang et al., 2021). To validate this choice, we compare Sim10K→Cityscapes adaptation performance with both objectives on top of a standard mixing objective. We find that cycle confusion underperforms against our proposed cycle consistency objective by 1.7 (i.e., 65.5 versus 67.2), and we consequently adopt cycle consistency throughout. With additional importance reweighting, we find that mAP@50 with cycle consistency further improves to 68.1.

▷ **`CARE` is agnostic to the underlying feature alignment strategy.** Our primary contribution is not a particular domain adaptation approach, but rather a novel *framework* designed for supervised domain adaptation, which, unlike prior work, explicitly leverages target labels in all components – conditional feature alignment, class-reweighting, and class-conditional box-reweighting, leading to strong improvements.

To validate this, we present target mAP@50 for Sim10K→Cityscapes adaptation by swapping out only the feature alignment strategy from cycle consistency to class-and-box-conditional versions of two popularly used domain adaptations strategies: maximum mean discrepancy (MMD) Long et al. (2015) minimization, and domain adversarial learning (DAL) Ganin & Lempitsky (2015). We find that both of these still significantly improve performance over no additional feature alignment (**+2.9** with MMD, **+3.9** with DAL).

### 4.6 `CARE`: Fine-grained performance analysis

Using Synscapes→Cityscapes, we analyze content-specific metrics to demonstrate that `CARE` consistently outperforms baselines in all settings and not just in aggregate.

▷ **`CARE` improves over baselines on all classes.** Fig. 6 (*left*) studies per-class performance improvements with our proposed method against baselines. Our method outperforms each baseline for every class.

▷ **Fine-grained error analysis.** We use TIDE (Bolya et al., 2020) to evaluate specific error types of our mixing baseline and `CARE` models (lower is better). Fig. 6 (*right*) shows that `CARE` reduces classification, localization, and duplicate errors, while slightly worsening joint classification+localization errors.

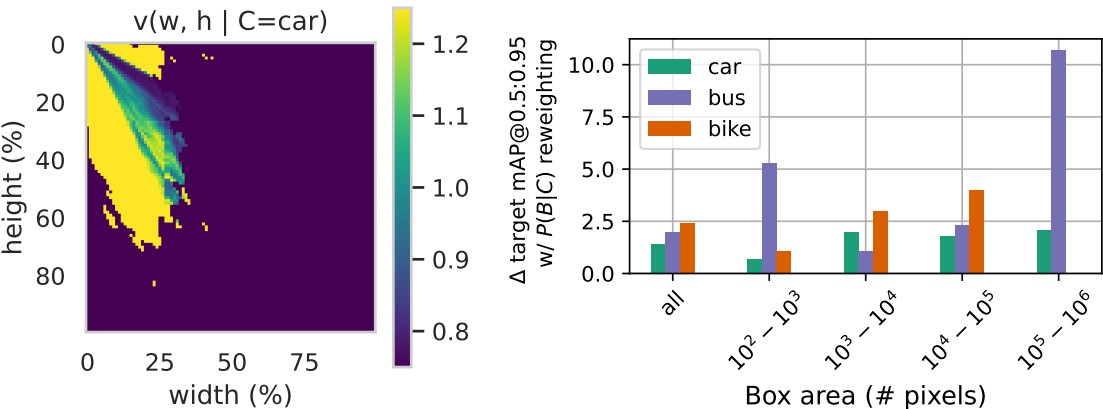

Figure 7: Visualizing $P(\mathsf{w},\mathsf{h}|C)$ reweighting on Synscapes→Cityscapes. **(left)** Visualizing $v(\mathsf{w},\mathsf{h}|C = \text{car})$. **(right)** Visualizing change in mAP after $P(\mathsf{w},\mathsf{h}|C)$ reweighting for three categories (car, bus, bike).

▷ **CARE improves per-class performance across box sizes.** Fig. 7 (*left*) illustrates bounding box frequency ratio weights $v(\mathsf{w},\mathsf{h}|C)$ for the "car" class estimated via the first term of Eq. equation 4. Matching our intuition (see Fig. 2, *right*), these ratios up-weight target cars of sizes that are relatively less frequent in the source domain. Fig. 7 (*right*) illustrates the change in mAP as a result of our reweighing for three categories over boxes of different sizes. Here, reweighing consistently improves mAP and can yield up to +10 mAP improvement for large objects such as buses. We these trends to hold for the remaining categories.

▷ **Visualizing matching with cycle consistency.** Fig. 3 provides a qualitative visualization of the matching behavior of our proposed cycle consistency approach, for two pairs of source and target images. For each example, we estimate the Euclidean distance in feature space between all cross-domain instance pairs in the aligned feature space of our `CARE` model and visualize the closest pair of car instances for each example. As expected, we find that our method embeds similar-looking cars closer in feature space.

## 5  Discussion

We study supervised Sim2Real adaptation applied to object detection and propose a strategy that exploits target labels to explicitly estimate and bridge the sim2real appearance and content gaps. Our method possesses a clear theoretical intuition and our empirical analyses validate our improvements in every setting that we tested, for example by boosting mAP@50 by as much as ∼25%. Most importantly, this paper tackles a large research-practice gap by bridging the literature on unsupervised and few-shot domain adaptation with an industry-standard practice of combining labeled data from both simulated and real domains. With this, we envision a renewed future methodological interest in SDA.

**Limitations.** Despite its strong performance, our method makes a few assumptions: First, it requires sufficient labeled data in source and target domains to reliably estimate dataset-level statistics. Further, our formulation assumes conditional independence of box sizes and locations as well as an equivalence between pixel-level and feature-level distributions. We also rely on successful cross-domain alignment. These assumptions may be violated to varying degrees in practice. Further, we do not consider an unlabeled portion of the target domain and leave that exploration to future work.

Prior work in visual domain adaptation primarily focuses on the tasks of image classification, semantic segmentation, and object detection. Of these, we focus on 2D detection as it is the most complex and practically applicable (for autonomous driving applications) of the three. Further, we explicitly address this additional complexity in our approach, e.g. by correcting for the sim2real shift in class-conditional size and location distributions using importance reweighting. However, while in theory such reweighting should be directly extensible to related tasks (*e.g.* 3D object detection) and settings (*e.g.* general transfer learning), such experiments are outside the scope of this paper.

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

## A  Appendix

### A.1  Varying mixing ratio

The mixing ratio controls the proportion of source and target data that are contained within a mini-batch of fixed size. In Fig. 8a, we vary the mixing ratio of real: simulated data and measure the subsequent performance on the Cityscapes test set. We use all source and target data for these experiments. We observed that the performance is fairly stable beyond a ratio of 50% and, for simplicity, we adopt this mixing ratio for all experiments unless otherwise stated.

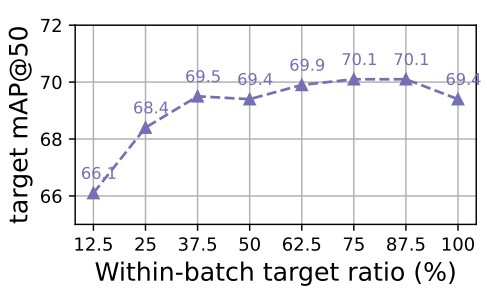 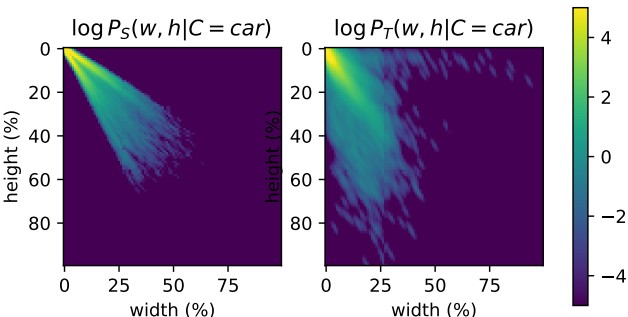

(a) Sim10K→Cityscapes (100% target data): Varying within-batch real: sim ratio for mixing.

(b) Visualizing log PDF values of KDE densities fitted to bounding box size on Synscapes→Cityscapes for the "car" class.

Figure 8: . Additional empirical analysis of `CARE`.

### A.2 Additional details on importance reweighting

When applying bounding-box importance reweighting we introduce a smoothing mechanism to ensure bounded loss values and thresholding to handle areas with low target support. Specifically, we compute:

$$v\left(B \mid C\right) = \begin{cases} \alpha\sigma\left(\frac{P_T(B|C)}{P_S(B|C)}\right) + \beta & \text{if } P_T\left(B \mid C\right) > \tau \\ 1.0 & \text{otherwise} \end{cases}$$

where $\alpha, \beta$ are scaling parameters (that we set to 20, -9, effectively bounding loss weights between 1 and 11). $\sigma$ denotes the sigmoid operator, and $\tau$ is a probability threshold that we set to 0.1. For boxes with very small target support, we simply set weights to a floor value of 1.0.

### A.3 Additional `CARE` analysis

**Visualizing KDE estimates.** In Figure 8b, we visualize log PDF values from KDE estimates fit to bounding box width and heights on both the source (Synscapes) and target (Cityscapes) domains. As seen, the KDE is able to capture the difference in box size distributions across domains: car class sizes vary significantly more in the target domain, consistent with our observation in Fig. 2. It is intuitive that with appropriate importance reweighting for source boxes as proposed in Sec. 3, we can improve overall performance across categories.

**Complexity analysis.** `CARE` does not add a substantial performance overhead over baselines. In fact, it is more efficient than sequential finetuning: we run CARE for 10k iterations, while sequential finetuning requires 20k (10k iterations for pretraining on source and 10k for finetuning on target). On the other hand, `CARE` and mixing have similar computational efficiency, with CARE being slightly slower due to additional class-conditional feature alignment, which necessitates learning an additional shallow encoder. Finally, we note that the per-class weights and class-conditional box weights are precomputed offline and so do not add a significant training-time overhead. Concretely, we find that training `CARE` for Sim10K→Cityscapes adaptation on 8 NVIDIA V100 GPUs with a per-GPU batch size of 4 takes approximately 3.5 hours to complete, as compared to 3 hours and 10 minutes for mixing.

### A.4 Additional Implementation Details

**Faster-RCNN training recipe**. For data augmentation, we only use random flipping (with 50% probability). For optimization, we use SGD with a learning rate of 0.01, a momentum of 0.9, and weight decay of $10^{-4}$. We train for 10k iterations and use a linear learning rate warmup for 500 iterations with a warmup ratio of 0.001. Following prior work (Wang et al., 2021), we decay the learning rate by a factor of 10 at 6k and 8k iterations each.

**Selecting $\lambda$.** We follow prior work (Prabhu et al., 2021) and tune loss weights ($\lambda$ in Equation 2) such that the feature alignment loss is of the same order of magnitude (after scaling) as the detection losses; we accordingly set $\lambda = 0.1$ across experiments.

