# OpenReview forum: "Bridging the Sim2Real gap with CARE: Supervised Detection Adaptation with Conditional Alignment and Reweighting"
_TMLR — Accepted by TMLR_

### Review · Reviewer_Hv1a · 2023-07-11

**Summary Of Contributions:**

The authors study the task of Sim2Real Domain Adaptation in a novel scenario where limited labeled target data is available. This is a reasonable setting in sensitive real-world applications. The authors use simulated, machine-labeled source data, and real, human-labeled target data, and aim to combine both in the most effective way. They characterize the domain gap between the source and target domains via the appearance gap and the content gap which they treat and address separately. The authors show superior results on all studied datasets compared to the baseline methods.

**Audience:**

Yes

**Broader Impact Concerns:**

I do not have concerns with regards to ethical implications of this work.

**Claims And Evidence:**

Yes

**Requested Changes:**

How was lambda in Eq. 2 set? Are the results sensitive to this hyperparameter? Please specify the model selection process here.

Could you please provide a complexity analysis of your method compared to other methods? For example, sequential finetuning likely requires more gradient steps because the model is first pretrained on the source data and then finetuned on the target data, so CARE should be more efficient? How efficient is CARE compared to the other baselines?

The authors study Supervised Domain Adaptation which, as they write, has not been studied much in the field of general Domain Adaptation since Unsupervised Domain Adaptation has been more popular. The setting they study is actually closer to transfer learning where a base model is pretrained on some large dataset (e.g. ImageNet) and then finetuned on a target task, e.g. on VTAB (https://ai.googleblog.com/2019/11/the-visual-task-adaptation-benchmark.html). Could you please establish a connection to transfer learning in the related work section? I think that testing CARE in a transfer learning setting is out of scope for this work since the paper already provides very detailed results in the studied setting of SDA. I think the presented results are definitely sufficient for this publication since the claims are well-supported by the presented evidence. Do the authors think that testing their approach in transfer learning could be interesting for future work? If so, could they maybe add it in the discussion section? My thought is that if this method has not been studied in transfer learning before, then maybe this paper would also be interesting to researchers working on transfer learning.  The way that the paper is currently written, I think the target audience will be researchers working on domain adaptation (but not transfer learning).

Question rather than a requested change: You write in the discussion: “First, it requires sufficient labeled data in source and target domains to reliably estimate dataset-level statistics.” Here, you refer to estimating the content and appearance statistics I assume. Could you please comment on aligning BatchNorm statistics between source and target and how CARE helps there? It has been shown previously (e.g. in [1]) that having separate BN layers per domain can be beneficial. Do you think CARE is implicitly aligning BN statistics of the source and target domains on the class / instance level?

[1] Rebuffi et al. “Learning multiple visual domains with residual adapters”


Related to the point above, could you formulate a stronger limitation for CARE (or maybe a minimal set of necessary requirements), for example if the source and target datasets are sufficiently different? In order to successfully reweigh the class probabilities, do all classes need to be present in both domains or can one domain have classes not present in the other? Do the target classes need to be a subset of the source classes? Is there a similar limitation for the cross-domain cycle consistency objective? Again, going back to transfer learning, using a pretrained model is almost always beneficial over training from scratch on a small labeled target dataset even if the tasks are very different. Is this the same for CARE?


**Strengths And Weaknesses:**

The paper is very well written and easy to follow. The authors motivate their object of study well and their approach is clear. On multiple occasions, the authors provide intuitions for their design choices which I appreciate. The Figures and Tables are very clear and well designed. The paper is well structured.

I appreciate that the authors include simple baselines which are not necessarily native to domain adaptation, such as mixing or sequential finetuning of the model.

The results are presented in a very detailed way. I especially appreciate Figure 5 where different ratios of the available target data are studied. I also appreciate Fig. 6, where the authors evaluate per-class performance, thereby checking that the improvements due to CARE are consistent across all classes.

Overall, the experiments have been performed carefully and the presented analyses are solid and detailed. I have some questions and suggestions for the authors which I outlined below, but did not discover any major flaws in this paper.

---

> ### Author Response · Authors · 2023-07-22
> **Thank you for the thoughtful review! (1/2)**
>
> Thanks for the thorough review! We’re delighted that the reviewer found this paper to be well-structured, our approach clear and intuitive, and our experimental section carefully designed and well-performed. We respond to specific questions below:
>
> > How was lambda in Eq. 2 set? Are the results sensitive to this hyperparameter? Please specify the model selection process here.
>
> We follow prior work [1] and tune loss weights such that the feature alignment loss is of the same order of magnitude (after scaling) as the detection losses, via grid search. We do not find performance to be particularly sensitive to $\lambda$, for instance, for Sim10K$\to$Cityscapes adaptation, we observe a target mAP@50 of 68.1 ($\lambda$=0.1) and 67.8 ($\lambda$=0.01)
>
> [1] Prabhu, Viraj, et al. "Sentry: Selective entropy optimization via committee consistency for unsupervised domain adaptation." Proceedings of the IEEE/CVF International Conference on Computer Vision. 2021.
>
> > Could you please provide a complexity analysis of your method compared to other methods? How efficient is CARE compared to the other baselines?
>
> The reviewer is correct that CARE is more efficient than sequential finetuning: we run CARE for 10k iterations, while sequential finetuning requires 20k (10k iterations for pretraining on source and 10k for finetuning on target). CARE and mixing have similar computational efficiency, with CARE being slightly slower due to additional class-conditional feature alignment; note that per-class weights $w(C)$ and class-conditional box weights $v(B|C)$ are precomputed offline, and so do not introduce an additional computational overhead during training. We will include this discussion in the paper, thanks for the suggestion!
>
> >Could you please establish a connection to transfer learning in the related work section?
>
> Transfer learning settings like the one studied in the VTAB benchmark are slightly more general in that they do not assume any task label set overlap between train and test distributions, whereas domain adaptation typically assumes a large (if not perfect) label set overlap. As a result, while both are concerned with learning _generalizable_ visual representations, domain adaptation methods typically attempt to also _align_ cross-domain feature representations (eg. via cross-domain cycle consistency feature alignment as done in CARE) to adapt transfer source representations to the target, whereas transfer learning methods typically do not. We would be happy to include this discussion in the related work and discussion sections of our revised draft, thanks!
>
> > "First, it requires sufficient labeled data in source and target domains to reliably estimate dataset-level statistics." Here, you refer to estimating the content and appearance statistics I assume. Could you please comment on aligning BatchNorm statistics between source and target and how CARE helps there? ... Do you think CARE is implicitly aligning BN statistics of the source and target domains on the class / instance level?
>
> By “sufficient labeled data ... to reliably estimate dataset-level statistics”, we were specifically referring to measuring task-level (class distribution) and scene-level (class-conditional box size and location distributions) content statistics. The unsupervised, semi-supervised, and few-shot DA settings assume no or very limited labeled target data, which precludes the reliable estimation of such statistics. In the label-privileged supervised DA setting we study, such estimation is possible, and as our experiments demonstrate, effective at bridging the content gap.
>
> As the reviewer points out, domain-specific batch-norm has indeed been shown to reduce cross-domain appearance discrepancies, and it is possible that CARE’s conditional feature alignment strategy aligns batch-norm statistics to some degree. However, since we update all model parameters rather than batch-norm alone, this lends the feature alignment with greater expressive power, to close say instance-level appearance gaps as well. Further, in addition to the appearance gap, CARE also address the content gap arising from changes in scene layout (which are unlikely to be captured by batch-norm statistics) via importance reweighting.

---

> > ### Author Response · Authors · 2023-07-22
> > **Thank you for the thoughtful review! (2/2)**
> >
> > > Related to the point above, could you formulate a stronger limitation for CARE (or maybe a minimal set of necessary requirements), for example if the source and target datasets are sufficiently different?
> >
> > We discuss some limitations of our method in Section 5, absent which CARE may not succeed:
> >
> > (1) CARE assumes sufficient labeled data in both source and target domain, without which the estimated scene-level statistics may not be reliable
> >
> > (2) CARE relies on an equivalence between pixel-level and feature-level alignment (Eq. 8). If the limit of very large domain discrepancy of insufficient model capacity, this may be violated.
> >
> > (3) CARE makes a simplifying assumption of conditional independence between box sizes and locations, which may not always hold, eg. near an image corner location x’, y’, all box sizes (w, h) are not equally likely.
> >
> > (4) CARE’s importance reweighting strategy assumes that target support is a subset of that of the source.
> >
> > > In order to successfully reweigh the class probabilities, do all classes need to be present in both domains or can one domain have classes not present in the other? Do the target classes need to be a subset of the source classes?
> >
> > As per limitation 4, the target support theoretically needs to be a subset of that of the source for importance reweighting to succeed. Otherwise, the ratio in Eq 4 ($\frac{P_T(B|C)}{P_S(B|C)}$) could be unbounded and lead to unstable training. In practice, this can be easily addressed by rescaling the loss weights as we describe in Sec A.2, or via techniques such as gradient clipping. In fact, in the Synscapes $\to$ Cityscapes shift, the `rider’ class is completely absent from the source domain, whereas the relative frequency of some other categories (e.g. car) is actually significantly higher in the target than in the source. Despite this theoretical violation of limitation 4, we find that in practice CARE still leads to consistent performance improvements on both these categories (see Figure 6).
> >
> > > Is there a similar limitation for the cross-domain cycle consistency objective?
> >
> > No, this limitation does not apply to the cross-domain cycle consistency objective. If a class is only present in one of the two domains, CARE will simply not perform feature alignment for that class.
> >
> > > Again, going back to transfer learning, using a pretrained model is almost always beneficial over training from scratch on a small labeled target dataset even if the tasks are very different. Is this the same for CARE?
> >
> > Absolutely! Consider Figure 5 (left), which studies the scaling behavior of target only training and CARE on the Sim10K to Cityscapes shift, in which the source domain (Sim10K) has relatively low photorealism: we restate the result in the table below for convenience, and denote in parenthesesthe performance improvement of CARE over target only training. Across label budgets, target only (training from scratch on target data) always lags behind CARE, which uses both source and target data. However, the gap reduces substantially as the amount of labeled target data increases (eg. from 10.2 points mAP@50 with 6.25% of target data labeled, to 1.5% with 100% of target data labeled).
> >
> > |  Sim10K    | 6.25%    | 12.5%  |  25%   |   50%      |   75%    | 100%    |
> > | ----------- | ----------- |-------------- | ----------- |-----------| --------------|--------
> > | Target    |   53.2      | 57.9 	| 65.2    | 66.2     |   68.1  | 69.4 |
> > | CARE     |   63.4 (+10.2)     |	 66.3 (+8.4) | 68.1 (+2.9) | 69.6  (+3.4)  | 70.9  (+2.7)  |  70.9 (+1.5)
> >
> > We would be happy to address any additional concerns, and will incorporate all other feedback into our revised draft.

---

> > > ### Comment · Reviewer_Hv1a · 2023-08-02
> > > **Response to the rebuttal**
> > >
> > > Dear authors,
> > >
> > > thank you for your detailed response to my review. The questions I had have now been resolved. I would appreciate it if, in the camera-ready version, you could include a) a detailed complexity analysis of CARE vs the other baselines, b) a discussion comparing the studied setting to transfer learning such e.g. fine-tuning of a pretrained model on VTAB, and c) details on how hyperparameters (such as lambda in eq. 2) were set; possibly, there are more (crucial) hyperparameter choices I missed and they should be documented.
> > >
> > > I have also read the other reviews and the authors' responses. I appreciate the experimental results on more advanced architectures and think that it further strengthens the provided empirical evidence. Please include those in the camera-ready version as well.
> > >
> > > Best,
> > > Reviewer Hv1a

---

> > > > ### Author Response · Authors · 2023-08-04
> > > > **Thank you for the excellent suggestions**
> > > >
> > > > We're glad that our response addresses your concerns, and have revised our draft as recommended (changes in blue). Please let us know if there are any other changes you would like us to incorporate!

---

### Review · Reviewer_jEpn · 2023-07-14

**Summary Of Contributions:**

This work proposes to study a new setting for domain adaptation in which the task is to maximize performance on a target domain having available some annotated data from the target domain and a lot  more from a different source domain. This is similar to the semi-supervised domain adaptation methods with the main difference being the availability of many annotated images from the target domain rather than only a few. The authors claim that these settings are relevant in industry especially in the context of autonomous driving where many real (target domain) and synthetic (source domain) data are available. The practical task studied by the authors is object detection. Their proposal to fight domain shift is to use cross domain feature matching to fight appearance shift between the two domains and loss rebalancing to fight content shifts by trying to matcn labels and box distributions. With these two modifications their proposal is able to significantly improve performance wrt a baseline trained only on the target domain as well as against algorithms for unsupervised or semi-supervised domain adaptation.

**Audience:**

Yes

**Broader Impact Concerns:**

No concern.

**Claims And Evidence:**

Yes

**Requested Changes:**

Please explain better the part criticized in weakness a-c-d and if possible improve the experimental validation as mentioned in weakness b.

**Strengths And Weaknesses:**

## Strengths

* Clear explanation of the motivations behind the work and on why it is an interesting task to explore from an industrial perspective.

* I appreciated the effort in proposing meaningful baselines to compare against. Since the authors are proposing a new protocol for evaluation they had not only to come up with the evaluation of their own solution but also of several baselines which require adapting other proposed methods.

## Weakness

a. **Presentation**: I found section 3.3.1 not clear and the notation confusing. In particular it is not clear to me how the soft match is defined. If I understood correctly $\hat{f}^j_T$ is supposed to be the reconstruction of $f^j_S$ as a weighted combination of all features from domain $T$. If that’s correct shouldn’t it have the subscript $S$? I found the notation quite confusing.
Also what’s the reason to apply this alignment only with the cycle source->target->source and not also target->source->target? In your setting both domains should have enough annotated samples to compute this regularization in a meaningful way.

b. **Experimental validation**: The authors picked a quite old architecture for detection (FasterRCNN with a Resnet50) and an unusual target dataset (Cityscape, which is usually benchmarked in the context of segmentation). In these settings it is unclear if the performance achieved is competitive in absolute terms or if the baselines selected are weak. In particular it could be interesting to either benchmark on a dataset which already provides several submitted methods (e.g., KITTI) or to take more modern state of the art architectures for detection and train them to the best of their possibility on the full target datasets (e.g., DETR [1] or ViDT [2]).

c. **Lack of details on the experimental evaluation**: from the current paper it is not clear why depending on the choice of source dataset the authors decide to test only on a subset of the classes in the target dataset. Is it a choice imposed by the datasets having only a subset of classes in common? Also other details missing regard the training recipe used to train the detection model. In particular it will be important to discuss the data augmentation strategy used as it has been proven to have a great impact in obtaining stronger baselines.

d. **Experimental settings violates the motivations of the paper**: The motivation of the paper claims the need to study realistic settings closer to industry standards, however in the end the author's main evaluation (Tab. 2) report results using only 25% of the available target data, while Fig. 5 shows that when using 100% of the target data often the gap shrink significantly. Note that according to the motivation of the paper the 100% target data setting should be the focus of the work. Also I found it a bit hard to justify the need of a loss rebalancing to minimize the content gap between synthetic and real dataset. In the context of the paper it is needed because the authors are studying datasets proposed by independent authors in academia, but in real setting the synthetic dataset can be generate to cover the same distribution (or a distribution as close as possible) of the target dataset, therefore maybe making the need of a content alignment less important.

## References
1. Carion, Nicolas, et al. "End-to-end object detection with transformers." European conference on computer vision. Cham: Springer International Publishing, 2020.
2. Song, Hwanjun, et al. "Vidt: An efficient and effective fully transformer-based object detector." arXiv preprint arXiv:2110.03921 (2021).

---

> ### Author Response · Authors · 2023-07-29
> **Thank you for the thoughtful review (1/2)**
>
> We are glad the reviewer found our task interesting, clear, and well-motivated, our performance gains significant, and appreciated our effort in proposing meaningful baselines. We address specific concerns below:
>
> > Results with more standard datasets and modern state-of-the-art architectures for detection
>
> We respectfully disagree that Cityscapes is not a standard dataset for object detection adaptation. In fact, to our knowledge, it is the most common _sim2real_ shift studied in the domain adaptation for object detection literature [A, B, C]. Additionally, most prior works reporting on this shift use Faster-RCNN with a ResNet-50 backbone as the detector, and we follow suit for comparability.
>
> However, we agree with the reviewer that this is a somewhat old architecture, and so as recommended, we run CARE with two additional architectures:
>
> * Replacing the ResNet-50 backbone in Faster-RCNN with Swin-T [D]: Swin is a hierarchical vision transformer architecture that uses shifted window attention across non-overlapping local windows which limits computation complexity to be linear with respect to image size, which improves efficiency and in turn allows for modeling multi-scale imagery.
>
> * VitDet [E]: ViTDet employs a plain, non-hierarchical vision transformer backbone (initialized with MAE [F] pretraining) with a simple feature pyramid network and standard window attention with a few cross-window propagation blocks. Despite its simple design, ViTDet achieves detection performance comparable with the state-of-the-art.
>
> Note that both of these make use of Vision Transformer (ViT) backbones, and ViTDet additionally benefits from self-supervised pretraining with MAE [E]. Below, we report target mAP@50 for target only, mixing, and CARE on Synscapes$\to$Cityscapes adaptation.
>
> | Synscapes | FRCNN + RN50 | FRCNN + Swin [A] | VitDet [B] |
> | ----------- | ---------------- |---------------- |---------------- |
> | Target    |   34.2    	|	44.8	|	 42.6  	|
> | Mixing    |  39.0     	|	44.9	|	 37.9  	|
> | CARE     | **48.5** 	|	**53.5**	| 	  **54.9** 	|
>
> As seen, even with recent vision transformer-based backbones and detectors CARE provides consistent and strong improvements over the baseline (eg. absolute improvement of +8.6 with Swin-T, +17 w/ VitDet, over mixing!). We will include these experiments in the main paper, thanks for the suggestion!
>
> [A] Yuhua Chen et al., Domain Adaptive Faster R-CNN for Object Detection in the Wild, CVPR 2018
>
> [B] Xin Wang et al., Robust Object Detection via Instance-Level Temporal Cycle Confusion, ICCV 2021
>
> [C] Yu-Jhe Li et al., Cross-Domain Adaptive Teacher for Object Detection, CVPR 2022
>
> [D] Ze Liu et al., “Swin transformer: Hierarchical vision transformer using shifted windows”, ICCV 2021
>
> [E] Yanghao Li et al., "Exploring plain vision transformer backbones for object detection.", ECCV 2022
>
> [F] Kaiming He et al., “Masked Autoencoders are Scalable Vision Learners”, CVPR 2022
>
> > Experimental setting violates the motivations of the paper.
>
> We focus on the 25% regime to mirror the real-life scenario of having an order of magnitude more labeled simulated images (_e.g._ 10k for Sim10K) than labeled real data (25%=625 for Cityscapes). In this setting, CARE clearly provides strong gains over baselines.
>
> As the reviewer points out, the improvements with CARE on this shift indeed shrink when 100% of the target dataset is labeled.  However, with a _larger_ source dataset Synscapes (25k images), we find that CARE again _significantly_ outperforms baselines even with 100% labeled target data (=2.5k for Cityscapes, an order of magnitude smaller than Synscapes). We include these results from the paper in the table below for convenience.
>
> |  Synscapes->Cityscapes (8-way)     | mAP@50  |
> | ----------- | ----------- |
> | Target    |   43.0     |
> | Mixing    |  46.9      |
> | Seq. FT  |    43.3   |
> | CARE     |    **51.9** |
>
> As Figure 5 of our paper shows, CARE provides consistent performance gains across target labeling budgets. However, it is most effective in the regime where we have an order of magnitude more simulated than real data. In industrial applications such as self-driving, this is easily accomplished with access to the simulator. We will clarify these details, thanks!

---

> > ### Author Response · Authors · 2023-07-29
> > **Thank you for the thoughtful review! (2/2)**
> >
> > > I found section 3.3.1 not clear and the notation confusing.
> >
> > Sorry about that! We have substantially revised 3.3.1 and hope that the explanation is clearer now. Let $\mathbf{u}^i$ denote features for the $i$-th source instance: in the forward pass, we first find features in the target domain to match it to, denoted by $\hat{\mathbf{v}}^i$, as an average of all target features $\mathbf{v}^j, j \in \{ 1, \ldots , k \}$, weighted by their pairwise similarity to $\mathbf{u}^i$. Matching to a weighted average of target features has been shown in prior work in cycle consistency to be more stable than matching to a single target instance [G].
> >
> > We agree that a the target$\to$source$\to$target cycle would be equally valid, and feasible to implement, but are not sure if it would provide a significant benefit in return for the additional computational overhead. We would be happy to incorporate additional suggestions to improve clarity!
> >
> > [G] Debidatta Dwibedi _et al._, “Temporal Cycle-Consistency Learning", CVPR 2019
> >
> > > Lack of details on the experimental evaluation.
> >
> > Yes, we report performance on the subset of categories that overlap across each source and target (Cityscapes) dataset pair: 1 for Sim10K (car), 8 for Synscapes, and 3 for Internal-Sim. We have added the requested training recipe details to appendix: to summarize, for data augmentation, we only use random flipping (with 50% probability). For optimization, we use SGD with a learning rate of 0.01, with a momentum of 0.9 and weight decay of $10^{-4}$ (Sec 4.2). We use a linear learning rate warmup for 500 iterations with a warmup ratio of 0.001, and decay learning rate by 10 at 6k and 8k iterations each. We train all models with  8 NVIDIA V100 GPUs with a per-GPU batch size of 4 and maintain a 1:1 ratio of source:target data across batches.
> >
> > > Also I found it a bit hard to justify the need of a loss rebalancing to minimize the content gap between synthetic and real dataset.
> >
> > Great question! While it is true that a source synthetic dataset can be generated so as to match the content distribution of the target, this would necessitate generating (and validating) a new dataset for each target deployment (say, in different geographies). In contrast, loss rebalancing can make use of a single synthetic dataset to adapt to a range of target datasets.

---

> > ### Comment · Reviewer_jEpn · 2023-08-02
> > **Response**
> >
> > Thanks for the details.
> >
> > I would suggest to include in the revised version of the paper:
> >
> > * The results with the new architectures
> > * Explicitly mention the relationship in size between sources and target datasets.
> > * Training recipe (also it would be great to try stronger augmentation than just flipping like: random rescaling, random cropping, color jittering, these can help significantly especially in the low data regime)
> > * Improved Section 3.3.1
> >
> > My doubts have been cleared.

---

> > > ### Author Response · Authors · 2023-08-04
> > > **Thank you for the excellent suggestions**
> > >
> > > We're glad that our response addresses your concerns, and have revised our draft as recommended (changes in blue). Please let us know if there are any other changes you would like us to incorporate.

---

### Review · Reviewer_YPSN · 2023-07-16

**Summary Of Contributions:**

This paper investigates the supervised sim2real DA problem with the 2D object detection task. To tackle this problem, the authors proposed a CARE method by minimizing both the appearance gaps and content gaps simultaneously. Specifically, the appearance gap is minimized by conditionally aligning the intermediate representations with ground-truth labels, while a reweighting scheme is proposed to overcome the content gap. Additionally, theoretical insights are provided. Experiments are conducted on the cityscape dataset.

**Audience:**

Yes

**Claims And Evidence:**

Yes

**Requested Changes:**

1. The effectiveness of the method can be validated more clearly with more advanced detection architecture.
2. More related references are suggested.

**Strengths And Weaknesses:**

Strength:
1. The investigated problem is well motivated, where labeled real and synthetic data are available in practice.
2. The appearance and content gaps are clearly presented and tackled with careful designs.
3. The proposed method is supported by theoretical analyses.
4. Improved performance over competitors, including UDA, FDA, and seq FT.

Weakness:
1. The proposed method is validated on a weak backbone of Faster-RCNN. It is interesting to present whether the proposed method is worked with more advanced backbone models.
2. Some missing references on DA, e.g., supervised DA is also investigated in (a), and (b) summarized various adversarial feature alignment methods.
    (a) Unified deep supervised domain adaptation and generalization, ICCV
    (b) Unsupervised multi-class domain adaptation: Theory, algorithms, and practice, TPAMI

---

> ### Author Response · Authors · 2023-07-29
> **Thank you for the thoughtful review!**
>
> We are delighted the reviewer found our problem well-motivated, our approach clearly presented and designed, and our empirical performance and theoretical justification convincing. We address specific concerns below:
>
> > The proposed method is validated on a weak backbone of Faster-RCNN. It is interesting to present whether the proposed method is worked with more advanced backbone models.
>
> As recommended, we ran CARE with two additional architectures:
>
> * Replacing the ResNet-50 backbone in Faster-RCNN to Swin-T [A]: Swin is a hierarchical vision transformer architecture that uses shifted window attention across non-overlapping local windows which limits computation complexity to be linear with respect to image size, which improves efficiency and in turn allows for modeling multi-scale imagery.
>
> * VitDet [B]: ViTDet employs a plain, non-hierarchical vision transformer backbone (initialized with MAE [C] pretraining) with a simple feature pyramid network and standard window attention with a few cross-window propagation blocks. Despite its simple design, ViTDet achieves detection performance competitive with state-of-the-art methods.
>
> Note that both of these make use of Vision Transformer (ViT) backbones, and ViTDet additionally benefits from self-supervised pretraining with MAE [C]. Below, we report target mAP@50 for target only training, mixing, and CARE on 8-way Synscapes$\to$Cityscapes adaptation. As seen, even with recent vision transformer-based backbones and detectors CARE provides consistent and strong improvements over baselines (eg. absolute gain of **+8.6** with Swin-T, **+17** w/ VitDet, over mixing!). We will include these experiments in the main paper, thanks for the suggestion!
>
> | Synscapes | FRCNN + RN50 | FRCNN + Swin [A] | VitDet [B] |
> | ----------- | ---------------- |---------------- |---------------- |
> | Target    |   34.2    	|	44.8	|	 42.6  	|
> | Mixing    |  39.0     	|	44.9	|	 37.9  	|
> | CARE     |  **48.5** 	|	**53.5**	| 	  **54.9** 	|
>
>
> [A] Ze Liu et al., “Swin transformer: Hierarchical vision transformer using shifted windows”, ICCV 2021
>
> [B] Yanghao Li et al., "Exploring plain vision transformer backbones for object detection.", ECCV 2022
>
> [C] Kaiming He et al., “Masked Autoencoders are Scalable Vision Learners”, CVPR 2022
>
> > Missing references on DA
>
> While Motiian _et al._ [D] also considers the supervised DA setting, it fundamentally differs from our work in two ways:
>
> i) Motiian _et al._ focuses on supervised DA for image classification and proposes bridging the cross-domain appearance gap by maximizing within-class cross-domain semantic alignment. We study supervised DA as applied to object detection, wherein we have to contend with an additional content gap, which we address with a novel importance reweighting strategy, in addition to a pixel and instance level appearance gap, which we minimize with class-conditional feature alignment based on cross-domain cycle consistency.
>
> ii) Motiian _et al._ focuses on the supervised DA regime wherein target data is sparingly labeled (eg. on Office-31, only 3 target examples / class are assumed to be labeled), whereas we assume a modest amount of labeled target data (in most of our experiments, ~625 Cityscapes images), which allows for reliable estimation of target statistics; our method explicitly leverages these statistics for bridging the aforementioned content gap.
>
> We have revised the related work section of the paper with these additional references, thanks for the suggestion!
>
> [D] Motiian et al, "Unified Deep Supervised Domain Adaptation and Generalization", ICCV 2017

---

### Decision · Action_Editors · 2023-08-02

**Recommendation:** Accept with minor revision

**Comment:**

All three reviewers found that the authors have solved their concerns and agree that the target problem is interesting and practical. The authors shall include the additional results (the results with two backbone architectures, ResNet-50+Swin-T and VitDet, and the results on Synscapes) in the rebuttal period and the requested information by reviewer Hv1a in the camera ready version, including
(a) a detailed complexity analysis of CARE vs the other baselines；
b) a discussion comparing the studied setting to transfer learning such e.g. fine-tuning of a pretrained model on VTAB；
c) details on how hyperparameters (such as lambda in eq. 2) were set).

**Audience:**

Yes.

**Claims And Evidence:**

Yes.